# Filtering walking actigraphy data in children with unilateral cerebral palsy: A preliminary study

Youngsub Hwang[1], Jeong-Yi Kwon[2]*

1 Department of Health Sciences and Technology, Samsung Advanced Institute for Health Sciences and Technology, Sungkyunkwan University, Seoul, Republic of Korea, 2 Department of Physical and Rehabilitation Medicine, Sungkyunkwan University School of Medicine, Samsung Medical Center, Seoul, Republic of Korea

* jeongyi.kwon@samsung.com

## Abstract

This study aimed to determine whether filtering out walking-related actigraphy data improves the reliability and accuracy of real-world upper extremity activity assessment in children with unilateral cerebral palsy. Twenty-two children aged 4–12 years diagnosed with unilateral cerebral palsy were included in this study, which was drawn from a two-phase randomized controlled trial conducted from July 2021 to December 2022. Data were collected from a tertiary hospital in Seoul, Republic of Korea. Participants were monitored using tri-axial accelerometers on both wrists across three time points (namely, T0, T1, and T2) over 3 days; interventions were used between each time point. Concurrently, an in-laboratory study focusing on walking and bimanual tasks was conducted with four participants. Data filtration resulted in a reduction of 8.20% in total data entry. With respect to reliability assessment, the intra-class correlation coefficients indicated enhanced consistency after filtration, with increased values for both the affected and less-affected sides. Before filtration, the magnitude counts for both sides showed varying tendencies, depending on the time points; however, they presented a consistent and stable trend after filtration. The findings of this research underscore the importance of accurately interpreting actigraphy measurements in children with unilateral cerebral palsy for targeted upper limb intervention by filtering walking-induced data.

## Introduction

Upper limb function is essential for various daily life activities [1]. The International Classification of Functioning, Disability, and Health (ICF) revolutionized the paradigm for assessing and treating children with cerebral palsy (CP) [2]. The ICF's biopsychosocial model, emphasizing both the physical and societal factors of disability, delineates the importance of optimizing every aspect of an individual's functioning [3]. Activities of daily living (ADL), which are pivotal for inclusion across school, home, and community environments, are categorized within the 'Activities and Participation' domain of the ICF and encompass fundamental life activities

**Data Availability Statement:** All relevant data are within the manuscript and its Supporting information files.

**Funding:** This research was made possible through the financial support of the National

Research Foundation of Korea (NRF) grant, which was provided by the Korean government (Grant No. 2021R1A6A3A13040170 and RS-2023-00278700).

**Competing interests:** The authors have declared that no competing interests exist.

such as bathing, eating, and performing chores [4]. Owing to their motor impairments and associated challenges, children with CP often diverge from the typical developmental trajectory, especially regarding ADL performance, which underscores the importance of understanding how these children function in real-world settings [5]. Consequently, an in-depth assessment of their actual upper extremity performance is vital to formulate evidence-based interventions that align with the ICF's holistic approach to health.

Actigraphy is a valuable tool that aids professionals in understanding the real-world performance patterns of children with neurodevelopmental disorders. By recording spontaneous and habitual movements in everyday settings, actigraphy can provide insights into a child's daily performance to complement traditional clinical assessments [6]. While conventional evaluations measure the individual's capacity [7], actigraphy may offer a window into real-world performance. Emphasizing the concurrent utilization of both these assessment approaches is crucial to bridge the gap between capacity and actual performance [8], ensuring a holistic understanding and more tailored therapeutic interventions for children with unilateral CP (UCP).

However, while valuable in capturing real-world performance [9, 10], actigraphy presents challenges in data interpretation [11], particularly distinguishing purposeful upper extremity actions from incidental movements, especially those generated during walking. Such incidental movements can obscure the data, leading to misrepresentation of a child's actual functional upper extremity use. Raw actigraphy data may not accurately reflect ADL performance without appropriate data filtration, posing risks in guiding therapeutic interventions. Therefore, refining actigraphy data to accurately capture intentional functional movements is essential.

To address this limitation, we examined actigraphy data filtration, emphasizing isolating walking-induced arm movements. Our primary goal was to determine the influence of refined data processing on the reliability and authenticity of actigraphy outcomes. We hypothesized that filtering out walking-associated data should enhance the consistency and reliability of actigraphy measurements for assessing upper extremity activity in children with UCP.

## Materials and methods

### Study design and participants

Children aged 4–12 years diagnosed with UCP due to lesions in the central nervous system participated in this nested study, which was drawn from a two-phase randomized controlled trial (RCT) conducted from July 4th, 2021 to December 22nd, 2022. Data were collected from a tertiary hospital in Seoul, Republic of Korea. The exclusion criteria included severe cognitive dysfunction that hindered the ability to perform simple tasks, untreated seizures, visual or auditory issues that could impede treatment, and a history of musculoskeletal disorders. The hospital review board approved the study (approval no. SMC 2021-04-042), and informed consent was acquired from the parents or legal guardians of the children before enrollment. The study was registered in the Clinical Trials database (clinical trial registration number: NCT04904796).

### Measures

**Physical activity monitoring using accelerometer-based monitors.** Physical activity information was collected using the ActiGraph wGT3X-BT (ActiGraph, Pensacola, FL), a 4.6×3.3×1.5-cm wireless, tri-axial accelerometer with a dynamic range of ±8 gravitational units at a sampling rate of 30 Hz. The selection of the ActiGraph wGT3X-BT model was underpinned by its documented reliability and validity in previous studies involving ambulatory individuals with CP [10, 12]. Vector magnitude average counts (VMA) and VMA ratios were

used as actigraphy variables. The VMA was calculated by adding the activity counts in all three axes, while the VMA ratio was determined by computing the natural logarithm (ln) of the ratio between the affected side VMA and less-affected side VMA (ln [affected side VMA/less-affected side VMA]). Acceleration data were downloaded and converted into 10-s epochs using ActiLife 6 software (ActiGraph, Pensacola, FL) and subsequently into activity counts.

## Procedures

**Data collection.**   The primary data collection encompassed 22 children with UCP, selected from a larger cohort of the main RCT. The participants wore accelerometers on their wrists for continuous physical activity monitoring over three consecutive days—specifically Friday, Saturday, and Sunday—to ensure uniformity across all measurement phases: baseline (T0), post-constraint-induced movement therapy (CIMT) (T1), and a 2-month follow-up (T2). Each day involved more than 10 h of monitoring using accelerometers affixed to both wrists. In parallel, supplementary data collection was undertaken in an in-laboratory setting with a small group of children with UCP. These children engaged in specified activities, encompassing walking and tasks necessitating the coordinated use of both hands. They were likewise outfitted with accelerometers on their wrists. Each in-laboratory session was delineated into a 10-min walking segment followed by a 30-min bimanual task performance segment. Bimanual tasks were carefully selected to represent everyday activities for children with UCP, including sitting and writing at a desk, using a tablet personal computer, playing board games, turning the pages of a book, and drawing. These activities, necessitating coordinated use of both hands, were distinctively compared with the more physically active task of walking.

**Activity classification and filtration in the test dataset.**   In this study, 'step count' refers to the number of steps recorded by the ActiGraph wGT3X-BT accelerometers. This metric was chosen as a key variable for its proven effectiveness in previous studies involving children with CP, where distinct activity patterns, including walking, were reliably identified by differing step count values [10, 12].

To classify and exclude walking-related data from the dataset, we employed a model developed from step count information gathered during a detailed in-laboratory study with a selected group of children with UCP. The reference model was established by analyzing the step count distributions, obtained during in-laboratory sessions that included walking and bimanual task sessions. For the procedure, the step count data underwent a 1-min rolling average. The subsequent classification was achieved by comparing the processed data to established distributions, categorizing each datum as "walking" or "tasks involving both hands." A computational function was developed to ascertain the probability of the dataset aligning with either "walking" or "tasks involving both hands." Data from both limbs is needed to indicate walking for a segment to be designated as such. Subsequently, data segments from the main dataset identified as "walking" were systematically filtered, a process we herein refer to as step reduction.

## Data analysis

Intraclass correlation coefficients (ICCs) were applied to the data from T0, T1, and T2 to assess the consistency of the actigraphy measurements. Analyses were conducted for the VMA variables on both sides, before and after step filtering. The extent of data filtration, including the decrease in column count and its associated percentage reduction, was analyzed. Paired $t$-tests were employed to determine differences in the VMA and VMA ratio measurements before and after filtration across all assessment points: T0, T1, and T2. All statistical procedures were conducted using SAS 9.4 and R 3.5.0, with a significance level set at $p < 0.05$.

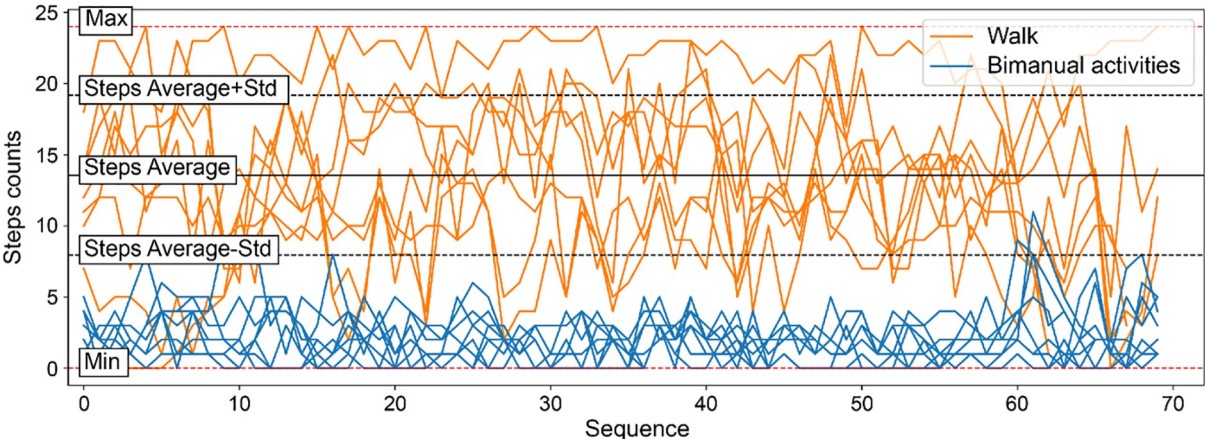

**Fig 1. Depiction of step patterns over time for walking and bimanual tasks.** The collective mean step count is demarcated by a continuous black line, with the standard deviation bounded by dashed black lines. Peak and trough step counts during ambulatory segments are signified by red dashed lines. Ambulatory activity is depicted in orange, symbolizing the dynamic nature of walking, while bimanual tasks are shown in blue, reflecting the controlled movements associated with these activities. This color-coded distinction facilitates a clear visual demarcation between the two types of activities, emphasizing the varying step counts associated with each. Std, standard deviation.

## Results

A total of 47 children with UCP were initially eligible for this study, of which 22 were included in the final analysis (12 boys; mean age: 5.48 ± 1.34 years); the rest were eliminated based on the exclusion criteria. All assessments were completed for each participant (S1 Fig). Detailed characteristics of the children are presented in S1 Table.

### Distinctive step patterns from in-laboratory walking and two-handed task performances

In a controlled in-laboratory setting, four children with UCP (children 7, 8, 12, and 22) exclusively engaged in two distinct activities: walking and tasks requiring the coordinated use of both hands. Based on the data obtained from these activities, we utilized the "Step counts" column to accurately differentiate between walking and two-handed task performances.

Fig 1 illustrates the step patterns derived from the "Step counts" column during walking and two-handed task performance for four participants.

S2 Fig focuses on the distribution of step counts during walking and two-handed tasks. This figure contrasts the frequency of specific step counts, offering insights into the variability and central tendencies of each activity type. Solid black lines indicate the average step count for walking and bimanual activities, with their respective variability represented by dashed lines for standard deviations. The color-coded presentation—walking in orange and bimanual tasks in blue—facilitates an intuitive understanding of density and distribution patterns, highlighting distinct characteristics of step counts associated with each type of activity.

### Magnitude of data filtration: A comparative analysis of pre- and post-filtering column counts

S2 Table presents the extent of column data filtration conducted over all time points (T0, T1, and T2), each spanning 3 days, focusing specifically on the total number of columns registered in 10-s epochs. The total columns for the entire cohort decreased from 800,594 (pre-filtering) to 734,924 (post-filtering), corresponding to a total reduction of 65,670 columns, with an

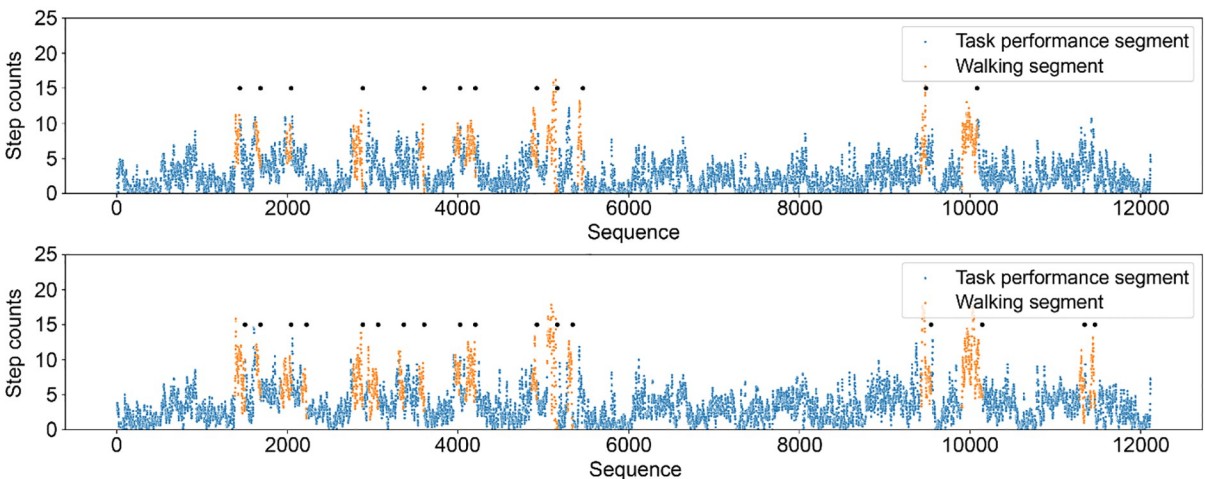

**Fig 2. A Comparative Visualization of Step Counts from One Participant's Actigraphy Data, Focusing on the Less-affected (Top) and Affected (Bottom) Sides.** Segments marked as 'Walking' in orange are identified for removal, streamlining the dataset for a focused analysis on 'Task performance.' This step is crucial in our methodology, as it ensures the precision of our assessment by excluding walking-induced steps that could obscure the evaluation of bimanual task performance.

average reduction rate of 8.20%. While individual children showed varied reduction rates ranging from as low as 3.07% to as high as 14.38%, the overall trend underscores the significant impact of the filtration process. Additionally, male participants had an average filtration of 9.48% (2.61), while female participants exhibited an average of 6.79% (2.42).

## Visualization of data filtration across participants' actigraphy

Fig 2 provides a detailed visualization of one participant's data, illustrating the granularity of how specific segments, such as "Task performance" and "Walking," were classified and filtered out. This visualization effectively demonstrates how walking-related data, depicted in orange, are isolated and excluded from the overall dataset to focus our analysis on non-ambulatory activities.

## Comparison of actigraphy values before and after filtration

The average of affected side VMA consistently displayed values of 774.10 (186.44) at T0, 797.49 (185.74) at T1 and 790.81 (171.28) at T2 in the non-filtered set. After filtration, these values changed to 720.85 (160.66) at T0, 716.40 (141.59) at T1, and 714.37 (137.36) at T2, elucidating the influence of walking-induced variables. The less-affected side's VMA revealed similar trends. In the non-filtered dataset, we observed values of 463.95 (145.56) at T0, 497.34 (156.20) at T1, and 487.26 (138.83) at T2. Upon filtering, these values were adjusted to 422.29 (122.41) at T0, 424.41 (117.46) at T1, and 418.46 (107.85) at T2. Intriguingly, across these variations in the VMA, the VMA ratio remained relatively steadfast for both the affected and less-affected sides. The average non-filtered VMA ratios were recorded as −2.34 (0.30) at T0, −2.36 (0.38) at T1, and −2.35 (0.45) at T2. After filtration, these ratios slightly adjusted to −2.32 (0.26) at T0, −2.35 (0.34) at T1, and −2.40 (0.41) at T2 (Table 1).

## Assessment of 3-day measurement consistency after step filtering

Data were gathered over 3 consecutive days at T0, T1, and T2 to assess the consistency and reliability of actigraphy measurements. The ICCs provide insight into the measurement

**Table 1. Comparison of non-filtered and walking-filtered variables over time with paired *t*-test results.**

| Variables | Non-filtered Mean (SD) | Filtered Mean (SD) | t | p |
|---|---|---|---|---|
| **T0 (days 1–3)** | | | | |
| Affected side VMA | 774.10 (186.44) | 720.85 (160.66) | 3.078 | **0.006**[a] |
| Less-affected side VMA | 463.95 (145.56) | 422.29 (122.41) | 3.593 | **0.002**[a] |
| VMA ratio | −2.34 (0.30) | −2.32 (0.26) | −0.740 | 0.468 |
| **T1 (days 1–3)** | | | | |
| Affected side VMA | 797.49 (185.74) | 716.40 (141.59) | 4.221 | <**0.001**[a] |
| Less-affected side VMA | 497.34 (156.20) | 424.41 (117.46) | 4.637 | <**0.001**[a] |
| VMA ratio | −2.36 (0.38) | −2.35 (0.34) | −0.402 | 0.692 |
| **T2 (days 1–3)** | | | | |
| Affected side VMA | 790.81 (171.28) | 714.37 (137.36) | 3.165 | **0.005**[a] |
| Less-affected side VMA | 487.26 (138.83) | 418.46 (107.85) | 3.887 | **0.001**[a] |
| VMA ratio | −2.40 (0.41) | −2.35 (0.45) | 1.215 | 0.238 |

[a]Significant difference.

VMA, vector magnitude average counts; SD, standard deviation.

reliability for both the affected and less-affected sides. The average values for the affected side VMA for T0, T1, and T2 were 0.876, 0.853, and 0.813, respectively. After step filtering, the ICC values increased to 0.888, 0.910, and 0.859. For the VMA of the less-affected side, the average values pre-filtering were 0.835, 0.839, and 0.828 for T0, T1, and T2, respectively. After filtering, the ICC values rose to 0.852, 0.865, and 0.861. The accompanying 95% confidence intervals further emphasized these findings (Table 2).

**Table 2. Intra-class correlation coefficients for the reliability of actigraphy measurements across 3 consecutive days at different evaluation points.**

| Measurements | Non-filtered | | Filtered[a] | |
|---|---|---|---|---|
| | ICC[a] | 95% CI | ICC[a] | 95% CI |
| **T0 (days 1–3)** | | | | |
| Affected side VMA | 0.876 | 0.750–0.945 | 0.888 | 0.770–0.950 |
| Less-affected side VMA | 0.835 | 0.661–0.928 | 0.852 | 0.684–0.931 |
| **T1 (days 1–3)** | | | | |
| Affected side VMA | 0.853 | 0.705–0.934 | 0.910 | 0.818–0.960 |
| Less-affected side VMA | 0.839 | 0.676–0.928 | 0.865 | 0.728–0.939 |
| **T2 (days 1–3)** | | | | |
| Affected side VMA | 0.813 | 0.610–0.919 | 0.859 | 0.708–0.939 |
| Less-affected side VMA | 0.828 | 0.642–0.925 | 0.861 | 0.714–0.939 |

T0, T1, and T2 represent three distinct evaluation points, each consisting of 3 consecutive days of measurements on both the affected and less-affected sides of the children.

[a]The ICC analysis was conducted to evaluate the consistency between measurements taken across 3 consecutive days (day 1, day 2, day 3) for each evaluation point (T0, T1, T2).

T0: baseline, T1: post-constraint-induced movement therapy, T2: post-hand-arm bimanual intensive training (experimental group) or equivalent time duration (control group).

CI, confidence interval; ICC, intra-class coefficient.

## Discussion

After filtration, there was a noticeable rise in ICC values, emphasizing the improved reliability when filtering walking-induced data. Notably, before this filtration, several measurements, such as the VMA of the less-affected side at T0 and both VMA measurements at T1, were below the 0.850 ICC threshold. However, post-filtering, all measurements surpassed this threshold. This enhancement may stem from the gait variability in young children [13]. Notably, one study showed that children with CP exhibited increased trunk sway velocity, especially when walking with restricted arm swing combined with increased speed, compared to conventionally developing children [14]. The complexity of gait dynamics in children with CP [15–17] and the potential alterations in trunk kinematics may thus contribute to the observed variability in actigraphy data for these children. By filtering out these walking-induced variations, the data may better reflect the actual upper extremity function, explaining the increased ICC values. For example, the VMA of the less-affected side at T0 rose from 0.835 to 0.852 after filtering, while the VMA of the affected side at T1 increased from 0.853 to 0.910. Such findings highlight the significance of step filtering in research on children with UCP, enhancing the reliability of actigraphy readings.

After applying the walking-induced filter, the actigraphy data offered a different perspective from the unfiltered data. In the unfiltered data, we observed an increase in activity in both the affected and less-affected sides post-CIMT at T1. However, these apparent increases were not sustained when the data were adjusted for walking-related movements. This suggests that our initial measurements might have reflected some unintentional movements in addition to the deliberate use of the upper extremities. Although laboratory assessments often indicate functional enhancements after CIMT [18–20], Goodwin et al. revealed a notable return to baseline in everyday arm movement following the intervention [21], indicating a gap between clinic-based achievements and their transfer to day-to-day lives of individuals with UCP. The findings from our filtered actigraphy data support the notion of a persistent gap between clinical progress and its translation into everyday functionality for individuals with UCP, further corroborating the results of previous research by Goodwin et al. [21].

The difference between clinical progress post-CIMT and actual performance in everyday contexts may necessitate a reassessment of the available therapeutic approaches; a broader approach to intervention that considers the child's developmental trajectory—encompassing physical, cognitive, sensory, and psychosocial dimensions—might be more effective. Our findings are consistent with those of Kwon et al., who reported that capacity improvements in children with CP do not uniformly translate into better performance outcomes [22]. This discrepancy suggests that clinical progress may not manifest as enhanced day-to-day functioning. Thus, our study highlights the potential advantages of a task-specific, home-based occupational therapy approach. According to the principles of the Cognitive Orientation to daily Occupational Performance (CO-OP) model, interventions should be tailored to each child's distinct environment and activities, potentially leading to substantial and practical improvements in daily life [23, 24].

In this study, we focused on UCP within the realm of actigraphy research, specifically accounting for walking data. A significant reduction was observed in the data, from 800,594 columns before filtration to 734,924 columns after filtration, representing an average exclusion of 2,985 columns per participant attributed to walking, highlighting the nuanced differences that emerge when raw data are subjected to thorough refinement. Notably, each column in our dataset corresponds to approximately ten steps, leading to an exclusion of roughly 29,850 steps over 3 days. To provide a comparative perspective, developing boys aged 6–11 years average between 12,000 to 16,000 steps daily, while girls average 10,000 to 13,000 [25]. Young

individuals with CP partake in 13%–53% less daily physical activity than their conventionally developing counterparts [26]. Considering this context, our filtration, which excluded nearly 30,000 steps over 3 days, seems consistent with expectations.

## Limitations

Several factors should be considered when interpreting our findings. First, we included a total sample size of 22 individuals, which may not fully represent the broader population of children with UCP, potentially impacting the generalizability of our results. Second, while the primary data are based on real-world situations, the supplementary data were collected from a controlled environment with only four children. This approach might not encompass the diverse range of activities typically performed by this demographic. Third, the activity classification system, derived from the step count of this small group, may introduce biases in activity categorization. Our research primarily addressed sway movements resulting from walking, but other incidental movements were not considered. Though beneficial in capturing movement patterns, accelerometers can pose challenges such as potential misclassifications or the omission of subtle movements. Another critical aspect is the validation of our activity classification model. The lack of a comprehensive validation process, such as cross-validation with a larger sample or external datasets, means that the reliability and accuracy of our classifications could be subject to question. Given these considerations, future research in pediatric OT could leverage advancements in human activity recognition technology [27, 28]. By integrating these technologies, subsequent studies might achieve a more detailed interpretation of actigraphy data, allowing for predicting specific ADLs. Expanding the sample size, implementing diverse data collection methods, and ensuring rigorous validation processes will be crucial steps in enhancing the depth, accuracy, and relevance of future research findings in this field.

## Conclusions

This preliminary study highlights the importance of thorough data filtration in actigraphy research for children with UCP. Our primary objective was to understand the influence of walking-induced data filtration on actigraphy outcomes. Consistent with our hypothesis, after step filtering, we observed improvements in the reliability of our measurements. Our findings, although preliminary, indicate that refining walking-induced data can improve the reliability of actigraphy measurements, as evidenced by increased ICCs, which may contribute to a more focused analysis of upper extremity activities by mitigating the influence of extraneous movements. While initial observations indicated increased activity on the affected side post-CIMT intervention, filtered data provided a subtler perspective, emphasizing the need for careful data analysis in such studies. Drawing parallels with Goodwin et al. [21], our initial findings suggest that although therapies such as CIMT may lead to some clinical improvements, their real-world impact on daily physical activities in children with UCP is likely nuanced. Given the preliminary nature of our study, a more extensive and rigorous approach in future research will be crucial to extract definitive conclusions on therapeutic outcomes.

## Supporting information

**S1 Table. Participant characteristics and neuroimaging findings.**
(DOCX)

**S2 Table. Column counts collected from T0 to T2 (each spanning 3 days) before and after filtering.**
(DOCX)

**S1 Fig. CONSORT 2010 flow diagram.**
(DOC)

**S2 Fig. Distribution of step counts for walking and tasks involving both hands.** The mean step count is represented by a solid black line, and the standard deviation range is shown using dashed black lines. Walking is represented in blue and bimanual tasks are shown in orange. Std, standard deviation.
(DOC)

## Acknowledgments

We are grateful to the research and clinical teams at the participating site, Samsung Medical Center (Seoul), for their contributions. Additionally, we wish to acknowledge the families who agreed to take part in this study.

## Author Contributions

**Conceptualization:** Youngsub Hwang.

**Data curation:** Youngsub Hwang, Jeong-Yi Kwon.

**Formal analysis:** Youngsub Hwang.

**Funding acquisition:** Jeong-Yi Kwon.

**Investigation:** Youngsub Hwang.

**Methodology:** Youngsub Hwang.

**Project administration:** Youngsub Hwang, Jeong-Yi Kwon.

**Software:** Youngsub Hwang.

**Supervision:** Jeong-Yi Kwon.

**Visualization:** Youngsub Hwang.

**Writing – original draft:** Youngsub Hwang.

**Writing – review & editing:** Youngsub Hwang, Jeong-Yi Kwon.

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
