## [Decision Letter · Decision Letter 0]

24 Mar 2024

PONE-D-24-04940Filtering walking actigraphy data in children with unilateral cerebral palsy: A preliminary studyPLOS ONE

Dear Dr. Kwon,

Thank you for submitting your manuscript to PLOS ONE. After careful consideration, we feel that it has merit but does not fully meet PLOS ONE’s publication criteria as it currently stands. Therefore, we invite you to submit a revised version of the manuscript that addresses the points raised during the review process.

We look forward to receiving your revised manuscript.

Kind regards,

Ibrahim Sadek, Ph.D.

Academic Editor

PLOS ONE

“This research was made possible through the financial support of the National Research Foundation of Korea (NRF) grant, which was provided by the Korean government (Grant No. 2021R1A6A3A13040170 and RS-2023-00278700).”

Reviewers' comments:

Reviewer's Responses to Questions

**Comments to the Author**

1. Is the manuscript technically sound, and do the data support the conclusions?

Reviewer #1: Yes

Reviewer #2: Yes

2. Has the statistical analysis been performed appropriately and rigorously? 

Reviewer #1: I Don't Know

Reviewer #2: Yes

3. Have the authors made all data underlying the findings in their manuscript fully available?

Reviewer #1: Yes

Reviewer #2: Yes

4. Is the manuscript presented in an intelligible fashion and written in standard English?

Reviewer #1: Yes

Reviewer #2: Yes

5. Review Comments to the Author

Reviewer #1: In this article, the authors present a solution to the problem of gait induced arm-swing in upper extremity actigraphy in the real world. The data collection was conducted simultaneously along a randomized controlled trial. The approach yielded positive results that might allow better utilization of activity monitors for at-home outcome measurement. I have listed my concerns with the article below:

Major:

1. The methods section needs to be rewritten for better readability. Multiple parts are repeated and, hence, redundant. e.g., Page 5 Line 98: "Acceleration data were downloaded and converted into 10-s epochs using ActiLife 6 software (ActiGraph, Pensacola, FL)" and Page 6 Line 116: "After collection, the acceleration data were downloaded and converted into 10-s epochs using the ActiLife 6 software (ActiGraph, Pensacola, FL)."

2. It is mentioned in the Data Collection section that the subjects were monitored for 3 days (need to mentioned clearly that this occurred at home/in subject's free living environment). There is no mention of what kind of days (weekdays/weekends) the data collection occurred, and whether the days were consistent between T0, T1, T2. Activities at home can change between weekdays and weekends.

Minor:

1. The subject count is mentioned multiple times before the demographics are stated. Please describe the patient recruitment, demographics, and exclusions either in the methods section or the results section.

2.Page 9 line 187: what is filtration rate? Did you mean reduction rate?

Reviewer #2: Children with unilateral cerebral palsy (UCP) often face a range of challenges that can impact their daily lives and development. This paper provides a study on " Filtering walking actigraphy data in children with UCP ". Specifically, it focuses on UCP within the realm of actigraphy research as checks whether the filtering out walking based calligraphy the reliability of assessments for children with unilateral cerebral palsy. The authors have provided all the details about the experimental data and how the achieved results indicated the possible effect of accurate interpreting actigraphy measurements in children with UCP for targeting upper limb intervention by filtering walking-induced data.

The paper is well organized and the English language quality of the entire paper is really good. However, I do recommend updating few references with up-to-date ones (if applicable). I do also see that the authors can give more explanations or discussions on the provided tables or graphs and also the same for the supporting document which can include a visual interpretation of the results or better highlighting the most dominant findings of the work.

6. PLOS authors have the option to publish the peer review history of their article (what does this mean?). If published, this will include your full peer review and any attached files.

Reviewer #1: No

Reviewer #2: **Yes: **Mohamed Tahoun

---

## [Author Response · Author response to Decision Letter 0]

4 Apr 2024

Journal requirements

-> Thank you for your detailed feedback and guidance regarding our manuscript submission. We have carefully reviewed PLOS ONE's style requirements and ensured that our manuscript conforms to these guidelines.

-> Regarding the financial disclosure in our manuscript, we have included a clear statement about the role of the funders. We thank you for the opportunity to clarify this aspect of our research support and for your guidance in ensuring the transparency of our financial disclosure.

Point-by-point Response to Reviewer’s Comments

Reviewer 1

Comment 1: The methods section needs to be rewritten for better readability. Multiple parts are repeated and, hence, redundant. e.g., Page 5 Line 98: "Acceleration data were downloaded and converted into 10-s epochs using ActiLife 6 software (ActiGraph, Pensacola, FL)" and Page 6 Line 116: "After collection, the acceleration data were downloaded and converted into 10-s epochs using the ActiLife 6 software (ActiGraph, Pensacola, FL)."

Response 1: Thank you for your insightful comment and constructive feedback. We appreciate the opportunity to improve the clarity and readability of our presentation.

In response to your observations, we have carefully revised the entire Methods section to eliminate redundancy. This included addressing the repeated mention of the downloading and conversion of acceleration data, the ActiLife 6 software (ActiGraph, Pensacola, FL), the model name, and, importantly, the duration for which the study participants wore the ActiGraph accelerometers. These repetitions have been removed to ensure a more streamlined and focused narrative.

Furthermore, we have restructured the subsection titles to better reflect the content. The "Participants" subsection has been renamed to "Study Design and Participants" to highlight the inclusion of study design information within this section. 

Additionally, we have relocated specific details to sections where they are more contextually appropriate. The explanation of the choice of the ActiGraph wGT3X-BT model, initially placed under "Data Collection," has been moved to "Physical Activity Monitoring Using Accelerometer-Based Monitors." This relocation better situates the rationale for selecting this device within the context of its usage for physical activity monitoring.

Moreover, the detailed description of the bimanual tasks was moved from "Activity Classification and Exclusion in the Test Dataset" to "Data Collection." This adjustment ensures that information on the selection and purpose of these tasks is introduced at a more relevant point in the Methods section, enhancing coherence and the logical flow of information.

Revised manuscript details are provided below.

Revised Manuscript:

Study design and participants

Children aged 4–12 years diagnosed with UCP due to lesions in the central nervous system participated in this nested study, which was drawn from a two-phase randomized controlled trial (RCT) conducted from July 4th, 2021 to December 22nd, 2022. Data were collected from a tertiary hospital in Seoul, Republic of Korea. The exclusion criteria included severe cognitive dysfunction that hindered the ability to perform simple tasks, untreated seizures, visual or auditory issues that could impede treatment, and a history of musculoskeletal disorders. The hospital review board approved the study (approval no. SMC 2021-04-042), and informed consent was acquired from the parents or legal guardians of the children before enrollment. The study was registered in the Clinical Trials database (clinical trial registration number: NCT04904796).

Measures

Physical activity monitoring using accelerometer-based monitors

Physical activity information was collected using the ActiGraph wGT3X-BT (ActiGraph, Pensacola, FL), a 4.6×3.3×1.5-cm wireless, tri-axial accelerometer with a dynamic range of ±8 gravitational units at a sampling rate of 30 Hz. The choice of the ActiGraph wGT3X-BT model was underpinned by its documented reliability and validity in previous studies involving ambulatory individuals with CP [10,12]. Vector magnitude average counts (VMA) and VMA ratios were used as actigraphy variables. The VMA was calculated by adding the activity counts in all three axes, while the VMA ratio was determined by computing the natural logarithm (ln) of the ratio between the affected side VMA and less-affected side VMA (ln [affected side VMA/less-affected side VMA]). Acceleration data were downloaded and converted into 10-s epochs using ActiLife 6 software (ActiGraph, Pensacola, FL) and subsequently into activity counts.

Procedures

Data collection

The primary data collection encompassed children with UCP, selected from the larger cohort of the main RCT. The participants wore accelerometers on their wrists for continuous physical activity monitoring over three consecutive days—specifically Friday, Saturday, and Sunday—to ensure uniformity across all measurement phases: Baseline (T0), post-constraint-induced movement therapy (CIMT) (T1), and a 2-month follow-up (T2). Each day involved more than 10 h of monitoring using accelerometers affixed to both wrists. In parallel, supplementary data collection was undertaken in an in-laboratory setting with a small group of children with UCP. These children engaged in specified activities, encompassing walking and tasks necessitating the coordinated use of both hands. They were likewise outfitted with accelerometers on their wrists. Each in-laboratory session was delineated into a 10-min walking segment followed by a 30-min bimanual task performance segment. Bimanual tasks were carefully selected to represent everyday activities for children with UCP, including sitting and writing at a desk, using a tablet PC, playing board games, turning the pages of a book, and drawing. These activities, necessitating coordinated use of both hands, were distinctively contrasted against the more physically active task of walking. 

Activity classification and filtration in the test dataset

In this study, 'step count' refers to the number of steps recorded by accelerometers. This metric was chosen as a key variable for its proven effectiveness in previous studies involving children with CP, where distinct activity patterns, including walking, were reliably identified by differing step count values [10,12].

To classify and exclude walking-related data from the dataset, we employed a model developed from step count information gathered during a detailed in-laboratory study with a select group of children with UCP. The reference model was established by analyzing the step count distributions, obtained during in-laboratory sessions that included walking and bimanual task sessions. For the procedure, the step count data underwent a 1-min rolling average. The subsequent classification was achieved by comparing the processed data to established distributions, categorizing each datum as “walking” or “tasks involving both hands.” A computational function was developed to ascertain the probability of the dataset aligning with either “walking” or “tasks involving both hands.” Data from both limbs is needed to indicate walking for a segment to be designated as such. Subsequently, data segments from the main dataset identified as “walking” were systematically filtered, a process we herein refer to as step reduction.

Comment 2: It is mentioned in the Data Collection section that the subjects were monitored for 3 days (need to mentioned clearly that this occurred at home/in subject's free living environment). There is no mention of what kind of days (weekdays/weekends) the data collection occurred, and whether the days were consistent between T0, T1, T2. Activities at home can change between weekdays and weekends.

Response 2: Thank you for pointing this out. We have specified in the manuscript that the data collection spanned three consecutive days—Friday, Saturday, and Sunday—for each participant. We intentionally made this selection to capture a comprehensive range of activities typically undertaken by children during weekends, when they are likely not attending school. This scheduling was consistent across all measurement phases: Baseline (T0), post-constraint-induced movement therapy (CIMT) (T1), and a 2-month follow-up (T2), to maintain uniformity in data collection and to allow for a more accurate comparison of physical activity patterns over time.

Revised manuscript details are provided below.

Revised Manuscript:

Procedures

Data collection

The primary data collection encompassed children with UCP, selected from the larger cohort of the main RCT. The participants wore accelerometers on their wrists for continuous physical activity monitoring over three consecutive days—specifically Friday, Saturday, and Sunday—to ensure uniformity across all measurement phases: Baseline (T0), post-constraint-induced movement therapy (CIMT) (T1), and a 2-month follow-up (T2). Each day involved more than 10 h of monitoring using accelerometers affixed to both wrists. In parallel, supplementary data collection was undertaken in an in-laboratory setting with a small group of children with UCP. These children engaged in specified activities, encompassing walking and tasks necessitating the coordinated use of both hands. They were likewise outfitted with accelerometers on their wrists. Each in-laboratory session was delineated into a 10-min walking segment followed by a 30-min bimanual task performance segment.

 

Comment 3: The subject count is mentioned multiple times before the demographics are stated. Please describe the patient recruitment, demographics, and exclusions either in the methods section or the results section.

Response 3: Thank you for your constructive feedback. We have thoroughly reviewed our manuscript with a particular focus on addressing the concern over numerical redundancy within the Procedures section. In response, we have revised our approach to describing the data collection and activity classification processes, ensuring that direct mentions of specific numerical counts of participants involved in these stages are strategically avoided. This adjustment has been made to streamline the narrative and enhance readability, particularly by generalizing descriptions where specific numbers previously appeared, thus aligning with your suggestions for improving clarity in our presentation.

Revised manuscript details are provided below.

Revised Manuscript:

Procedures

Data collection

The primary data collection encompassed children with UCP, selected from the larger cohort of the main RCT. The participants wore accelerometers on their wrists for continuous physical activity monitoring over three consecutive days—specifically Friday, Saturday, and Sunday—to ensure uniformity across all measurement phases: Baseline (T0), post-constraint-induced movement therapy (CIMT) (T1), and a 2-month follow-up (T2). Each day involved more than 10 h of monitoring using accelerometers affixed to both wrists. In parallel, supplementary data collection was undertaken in an in-laboratory setting with a small group of children with UCP. These children engaged in specified activities, encompassing walking and tasks necessitating the coordinated use of both hands. They were likewise outfitted with accelerometers on their wrists. Each in-laboratory session was delineated into a 10-min walking segment followed by a 30-min bimanual task performance segment. Bimanual tasks were carefully selected to represent everyday activities for children with UCP, including sitting and writing at a desk, using a tablet PC, playing board games, turning the pages of a book, and drawing. These activities, necessitating coordinated use of both hands, were distinctively contrasted against the more physically active task of walking. 

Activity classification and filtration in the test dataset

In this study, 'step count' refers to the number of steps recorded by accelerometers. This metric was chosen as a key variable for its proven effectiveness in previous studies involving children with CP, where distinct activity patterns, including walking, were reliably identified by differing step count values [10,12].

To classify and exclude walking-related data from the dataset, we employed a model developed from step count information gathered during a detailed in-laboratory study with a select group of children with UCP. The reference model was established by analyzing the step count distributions, obtained during in-laboratory sessions that included walking and bimanual task sessions. For the procedure, the step count data underwent a 1-min rolling average. The subsequent classification was achieved by comparing the processed data to established distributions, categorizing each datum as “walking” or “tasks involving both hands.” A computational function was developed to ascertain the probability of the dataset aligning with either “walking” or “tasks involving both hands.” Data from both limbs is needed to indicate walking for a segment to be designated as such. Subsequently, data segments from the main dataset identified as “walking” were systematically filtered, a process we herein refer to as step reduction.

 

Comment 4: Page 9 line 187: what is filtration rate? Did you mean reduction rate?

In our manuscript, "filtration" is used interchangeably with "step reduction." We have clarified this in the manuscript, as follows: "Subsequently, data segments from the main dataset identified as 'walking' were systematically filtered, a process we herein refer to as step reduction." This choice was to ensure our terminology precisely reflects the process undertaken—systematic removal of walking-related data to isolate upper limb activity. 

Revised manuscript details are provided below.

Revised Manuscript:

Activity classification and filtration in the test dataset

In this study, 'step count' refers to the number of steps recorded by accelerometers. This metric was chosen as a key variable for its proven effectiveness in previous studies involving children with CP, where distinct activity patterns, including walking, were reliably identified by differing step count values [10,12].

To classify and exclude walking-related data from the dataset, we employed a model developed from step count information gathered during a detailed in-laboratory study with a select group of children with UCP. The reference model was established by analyzing the step count distributions, obtained during in-laboratory sessions that included walking and bimanual task sessions. For the procedure, the step count data underwent a 1-min rolling average. The subsequent classification was achieved by comparing the processed data to established distributions, categorizing each datum as “walking” or “tasks involving both hands.” A computational function was developed to ascertain the probability of the dataset aligning with either “walking” or “tasks involving both hands.” Data from both limbs is needed to indicate walking for a segment to be designated as such. Subsequently, data segments from the main dataset identified as “walking” were systematically filtered, a process we herein refer to as step reduction.

 

Reviewer 2

Comment 1: The paper is well organized and the English language quality of the entire paper is really good. However, I do recommend updating few references with up-to-date ones (if applicable). 

Response 1: Thank you for your insightful feedback and recommendation. Following your suggestion, we have carefully reviewed our reference list, changing a reference from the ‘80s to a more recent publication. 

Revised Manuscript:

1. Klingels, Katrijn, et al. "Upper limb impairments and their impact on activity measures in children with unilateral cerebral palsy." Eur J Paediatr Neurol. 2012;16(5):475-484.

 

Reviewer 2

Comment 2: I do also see that the authors can give more explanations or discussions on the provided tables or graphs and also the same for the supporting document which can include a visual interpretation of

---

## [Decision Letter · Decision Letter 1]

16 Apr 2024

Filtering walking actigraphy data in children with unilateral cerebral palsy: A preliminary study

PONE-D-24-04940R1

Dear Dr. Kwon,

We’re pleased to inform you that your manuscript has been judged scientifically suitable for publication and will be formally accepted for publication once it meets all outstanding technical requirements.

Kind regards,

Ibrahim Sadek, Ph.D.

Academic Editor

PLOS ONE

Additional Editor Comments (optional):

Reviewers' comments:

Reviewer's Responses to Questions

**Comments to the Author**

1. If the authors have adequately addressed your comments raised in a previous round of review and you feel that this manuscript is now acceptable for publication, you may indicate that here to bypass the “Comments to the Author” section, enter your conflict of interest statement in the “Confidential to Editor” section, and submit your "Accept" recommendation.

Reviewer #1: All comments have been addressed

2. Is the manuscript technically sound, and do the data support the conclusions?

Reviewer #1: Yes

3. Has the statistical analysis been performed appropriately and rigorously? 

Reviewer #1: I Don't Know

4. Have the authors made all data underlying the findings in their manuscript fully available?

Reviewer #1: Yes

5. Is the manuscript presented in an intelligible fashion and written in standard English?

Reviewer #1: Yes

6. Review Comments to the Author

Reviewer #1: (No Response)

7. PLOS authors have the option to publish the peer review history of their article (what does this mean?). If published, this will include your full peer review and any attached files.

Reviewer #1: No

---

## [Editor Report · Acceptance letter]

29 Apr 2024

PONE-D-24-04940R1 

PLOS ONE

Dear Dr. Kwon, 

I'm pleased to inform you that your manuscript has been deemed suitable for publication in PLOS ONE. Congratulations! Your manuscript is now being handed over to our production team.

Kind regards, 

on behalf of

Dr. Ibrahim Sadek 

Academic Editor

PLOS ONE